# Early Colorectal Responses to HIV-1 and Modulation by Antiretroviral Drugs

**DOI:** 10.3390/vaccines9030231

**Published:** 2021-03-08

**Authors:** Carolina Herrera, Mike D. McRaven, Ken G. Laing, Jayne Dennis, Thomas J. Hope, Robin J. Shattock

**Affiliations:** 1Centre for Infection & Immunity, Division of Clinical Sciences, St George’s University of London, London SW17 0RE, UK; r.shattock@imperial.ac.uk; 2Department of Cell and Molecular Biology, Feinberg School of Medicine, Northwestern University, Chicago, IL 60611, USA; m-mcraven@northwestern.edu (M.D.M.); thope@northwestern.edu (T.J.H.); 3Division of Clinical Sciences, St George’s University of London, London SW17 0RE, UK; klaing@sgul.ac.uk (K.G.L.); jayne.dennis@qmul.ac.uk (J.D.)

**Keywords:** HIV-1, mucosal tissue, antiretrovirals, pre-exposure prophylaxis

## Abstract

Innate responses during acute HIV infection correlate with disease progression and pathogenesis. However, limited information is available about the events occurring during the first hours of infection in the mucosal sites of transmission. With an ex vivo HIV-1 challenge model of human colorectal tissue we assessed the mucosal responses induced by R5- and X4-tropic HIV-1 isolates in the first 24 h of exposure. Microscopy studies demonstrated virus penetration of up to 39 μm into the lamina propia within 6 h of inoculation. A rapid, 6 h post-challenge, increase in the level of secretion of inflammatory cytokines, chemokines, interferon- γ (IFN-γ), and granulocyte-macrophage colony-stimulating factor (GM-CSF) was observed following exposure to R5- or X4-tropic isolates. This profile persisted at the later time point measured of 24 h. However, exposure to the X4-tropic isolate tested induced greater changes at the proteomic and transcriptomic levels than the R5-tropic. The X4-isolate induced greater levels of CCR5 ligands (RANTES, MIP-1α and MIP-1β) secretion than R5-HIV-1. Potential drugs candidates for colorectal microbicides, including entry, fusion or reverse transcriptase inhibitors demonstrated differential capacity to modulate these responses. Our findings indicate that in colorectal tissue, inflammatory responses and a Th1 cytokine profile are induced in the first 24 h following viral exposure.

## 1. Introduction

HIV prevention strategies are essential to curb the rate of new infections. Vaccine candidates have so far been designed to induce virus-specific antibody of T cell responses; however, animal studies and clinical trials have shown limited protection highlighting the need to better understand the links between innate and adaptive response(s) capable of mediating protection. Drug based prevention approaches have shown greater success with antiretroviral drugs (ARVs) such as Truvada, a combination of two nucleot(s)ide reverse transcriptase inhibitors (Tenofovir/Emtricitabine) [1,2,3], although lack of user adherence and mucosal factors such as inflammation and microbiota have been shown to hinder the efficacy of ARVs [4]. Hence, new approaches to HIV prophylaxis are urgently needed and understanding the innate responses induced in the first hours of exposure to HIV is essential to design such strategies.

Chronic inflammation observed during HIV-1 infection is well known to be associated with disease progression [5]. During the acute phase of infection, plasma viral titers have been shown to correlate with a rapid systemic increase of multiple cytokines (interferon-α (IFN-α), IFN-β, inducible protein 10 (IP-10), tumor necrosis factor (TNF), monocyte chemotactic protein 1 (MCP-1) interleukin-6 (IL-6), IL-8, IL-10, IL-15 and IL-18) in the first 25 days post-infection [6]. However, human studies are limited by the estimation of the infection time point. Furthermore, sparse information is available about the initial immune responses at the mucosal sites of transmission. Non-human primate studies with simian immunodeficiency virus indicate that transmission in the female genital tract (FGT) involves establishment of an initial focus of infections or ‘founder population’ in the first hours after viral exposure and increased secretion of IFN-α, IFN-β, macrophage inflammatory protein 1α (MIP-1α), MIP-1β, MIP-3α, and IL-8 [7,8].

Receptive anal intercourse (RAI) between serodiscordant couples in both men and women is associated with the highest probability of sexual HIV transmission [9,10,11,12] and the colorectal tract is the major site of viral replication and CD4^+^ T-cell depletion during acute infection [13]. Hence, in this study, we characterized the level of expression of the HIV receptor and coreceptors in colorectal tissue, and evaluated, at proteomic and transcriptomic levels, the immune responses induced in the first hours following ex vivo exposure of mucosal tissue to clade B R5- and X4-tropic HIV-1 isolates. Furthermore, we analyzed the potential effect of ARVs on the early mucosal immune responses induced by exposure to HIV-1.

## 2. Materials and Methods

### 2.1. Reagents

PMPA (tenofovir) was donated by Gilead Sciences, Inc. (Foster City, CA, USA), UC781 by Biosyn, Inc. (Huntington Valley, PA, USA), and TMC120 (dapivirine) by Janssen ID and V (Belgium). T1249, CMPD-167 and AMD3465 were donated by the International Partnership for Microbicides (IPM) (Silver Spring, MD). Drugs were used at noncytotoxic concentrations determined by a 3-(4,5-dimethyl-2-thiazolyl)-2,5-diphenyl-2H-tetrazolium bromide (MTT) viability assay.

### 2.2. Plasmids and Virus Culture Conditions

Full-length, replication and infection competent proviral HIV-1 clones pYU2 [14,15] and pNL4.3 [16] were provided by the NIH AIDS Research and Reference Reagent Program (http://www.aidsreagent.org/) (accessed on 8 May 2006). PA-GFP-Vpr-HIV-1 BaL was generated and characterized by Prof. TJ. Hope as described previously [17,18]. Briefly, PA-GFP-Vpr-labeled HIV-1 BaL was produced in two phases. A GFP-Vpr HIV-1 BaL construct was generated by transfection of 293T cells with a proviral HIV-1 BaL construct and the plasmid peGFPC3 (CLONTECH laboratories, Inc., Mountain View, CA, USA) containing the entire Vpr-coding region fused to the COOH terminus of eGFP. GFP was then substituted by PA-GFP [19].

Molecular clones were passaged through activated PBMCs for 11 days. PBMCs were isolated from multidonor buffy coats from healthy HIV-seronegative donors, by centrifugation onto Ficoll-Hypaque, mitogen stimulated as described previously [20], and maintained in RPMI 1640 medium containing 10% FCS, 2 mM L-glutamine, antibiotics (100 U of penicillin/mL, 100 µg of streptomycin/mL), and 100 U of interleukin-2/mL.

Virus was inactivated with 2,2′-dithiodipyridine (aldrithiol-2; AT-2) [21] for control experiments.

### 2.3. Patients and Tissue Explants

Surgically resected specimens of intestinal tissue were collected at St George’s Hospital, London, UK after receiving signed informed consent. All patients were HIV negative. All tissues were collected under protocols approved by the Local Research Ethics Committee. Following transport to the laboratory, tissue was dissected and cut into 2–3 mm^3^ or 1 cm^3^ explants comprising both epithelial and muscularis mucosae as described previously [22]. Colorectal explants were maintained with DMEM containing 10% fetal calf serum, 2 mM L-glutamine and antibiotics (100 U of penicillin/mL, 100 µg of streptomycin/mL, 80 µg of gentamicin/mL), at 37 °C in an atmosphere containing 5% CO_2_.

### 2.4. Tissue Digestion and Flow Cytometry Analysis

Colorectal tissue was digested with an enzyme cocktail including dispase I and DNase I, as previously described [23]. The resulting cellular suspension was washed in PBS, counted and stained for flow cytometry analysis at a concentration of 10^6^ cells/mL. Mononuclear cells (MNCs) were fixed in a PBS solution containing 2% paraformaldehyde, 60 mM sucrose at pH 7.4, for 15 min at room temperature. Fixation is necessary for MAb 45523 to bind to cell surface CCR5 on primary CD4^+^ T-cells. The fixed cells were washed twice with PBS containing 20 mM glycine (buffer A) and incubated for 15 min at room temperature in buffer A supplemented with 1% BSA and 0.05% NaN_3_ (buffer B). Cells were then stained with mixtures of MAbs in buffer B, anti-CD4-APC-Cy7, anti-CD13-PE, anti-CCR5-2D7-FITC, anti-CXCR4-12G5-FITC, anti-HLA-DR-PE-Cy7 (BD Pharmingen, San Diego, CA), anti-CCR5-45531-FITC, anti-CCR5-45523-FITC, anti-DC-SIGN-PE (R&D Systems), anti-CD14-ECD, anti-CD64-PE, or anti-89-PE (Beckman Coulter), for 1 h at room temperature. After three washes in buffer B, cells were resuspended for analysis using a BD LSR II flow cytometer (BD Biosciences; San Jose, CA, USA). The parameters used to select cell populations for analysis were forward and side-light scatter, with a total of 10,000 events being collected for analysis as described previously [24].

### 2.5. Infectivity and Drug Exposure Assays

The infectivity of virus preparations was estimated in PBMCs by measuring p24 antigen content in cell culture supernatant (HIV-1 p24 ELISA, AIDS Vaccine Program, National Cancer Institute, Frederick, MA, USA), as described previously [25].

Tissue explants were incubated in a nonpolarized system with or without inhibitory concentrations of drugs for 1 h at 37 °C before addition of virus (10^3^ TCID_50_) for 2 h. Explants were then washed four times with PBS before transferring to fresh plates. Explants were cultured for 6 or 24 h to assess early responses.

For virion visualization, 1 cm^3^ tissue explants were exposed in a nonpolarized system for 6 h to HIV-1BaL-PA-GFP-Vpr (10^3^ TCID_50_), then washed four times with PBS and snap-frozen at −80 °C, in optimal cutting temperature (OCT) compound. Tissue was sectioned (12 µm) for immunofluorescent analysis.

### 2.6. Immunohistochemistry

Frozen tissue sections (12 µm) were fixed in 3.7% formaldehyde in PIPES (piperazine-N,N’-bis(2-ethanesulfonic acid)) buffer for 5–10 min followed by washing in cold PBS. Samples were blocked with normal donkey serum for 10 min and washed again. For identification of intestinal columnar epithelial structure, samples were stained with cytokeratin-20 (clone OV-TL; DakoCytomation, Carpinteria, CA, USA) at room temperature (RT) for 1 h. To identify target cells, tissue sections were stained with anti-CD209 (clone 120507) (R&D Systems) for dendritic cells, anti-CD4-OKT4 (Cell Marque, Rocklin, CA) for CD4^+^ cells, and anti-CD68-EBM11 (DakoCytomation) for macrophages, each for 1 h at RT. To highlight cellular glycoproteins in the columnar epithelium, fluorescently tagged wheat germ agglutinin (WGA) (Invitrogen, Carlsbad, CA, USA) was added at RT for 30 min. Secondary antibodies (Jackson ImmunoResearch, West Grove, PA, USA) used were labeled with rhodamine RedX or Cyanine 5 (Cy5). Finally, Hoechst DAPI (4,6-diamidino-2-phenylindole) (Invitrogen) was applied for 10 min to stain nuclei before a final wash with PBS. Mounting medium (DakoCytomation) and coverslips were applied to sections, sealed and stored at 4 °C until imaged.

### 2.7. Imaging

Images were obtained by deconvolution microscopy on a DeltaVision RT system collected on a digital camera (CoolSNAP HQ; Photometrics, Tucson, AZ, USA). For virus penetration analyses, z-scan stacks were collected over 15 µm for each image field, and image analysis was performed with the softWoRx software (Applied Precision, Bratislava, Slovakia). To determine the extent of viral penetration, the distance from the epithelial surface to each virion was measured using the measuring capabilities of softWoRx software. Penetrators were defined as virions entering more than 1 µm into the epithelium. For each sample, approximately 20 z-scan stack images were randomly obtained to allow the interaction of virus with genital mucosa to be compared across a large surface area. The regions imaged were 60 µm wide and 12 µm thick.

### 2.8. Multiplex Cytokine Analysis

The level of 23 cytokines in tissue supernatants after 6 or 24 h of culture were quantified by in house multiplex bead immunoassay as described previously [26] using a Luminex 100 Systems (Bio-Rad, Hercules, CA, USA).

### 2.9. Microarray Analysis of Gene Expression

#### 2.9.1. Total RNA Extraction and Quality Control

Tissue explants harvested at 6 or 24 h were kept overnight in 200 µL of RNAlater reagent and then frozen at −80 °C prior to extraction of RNA. RNA stabilized tissue explants were transferred to Lysing Matrix A (MP Bio-medicals, Santa Ana, CA, USA) and homogenized with a FastPrep Ribolyser (MP Biomedicals). Lysed tissue supernatants were collected, and total RNA purified using RNeasy mini kit (Qiagen, Venlo, The Netherlands) following manufacturer’s instructions.

#### 2.9.2. RNA Microarray Analysis

RNA was reverse transcribed into cDNA, amplified and transcribed into cRNA using the Illumina TotalPrep RNA amplification kit (Ambion, Life Technologies, Warrington, UK) according to manufacturer’s instructions. The cRNA was then assessed for quantity and quality by Agilent RNA 6000 Pico Kit (Agilent Technologies, Germany) and NanoDrop (ThermoScientific, West Sussex, UK) and hybridized to Illumina BeadChip Array Single Color Human HT-12_v3_Beadchip (Illumina) according to manufacturer’s instructions and scanned on an Illumina GX500 Beadstation.

### 2.10. Statistical Analysis

Cytokine concentrations a were calculated from sigmoid curve-fits (Prism, GraphPad). All data presented fulfill the criterion of R^2^ > 0.7. Concentration values were statistically compared using unpaired *t* test and *p* values. Treatment conditions were considered significant when *p* < 0.05.

RNA microarray data was analyzed in GeneSpring v11.5.1 (Agilent Technologies). In brief, data was normalized using quantile normalization and baseline transformed to the median of all control samples. Data was filtered to remove unexpressed or unreliable data such that remaining entities should have a detection *p*-value of greater than 0.6 in 100% of samples for any one of the two conditions and a fold change of greater than 1.5 compared to the control. Entrez Gene, available in the NCBI homepage (www.ncbi.nlm.nih.gov/gene) (accessed on 22 June 2017), was used to identify all known gene ontology and the processes associated with the products of those genes. An unpaired *t*-test was performed in conjunction with a Benjamini–Hochberg multiple testing correction and a corrected *p* value of <0.05 was applied.

Protein and gene levels were normalized to the matched explant controls instead of the median of the control treatments to improve the correction for explant effects and log transformed (base 2). Statistical significance of differences between treated and untreated tissue explants were determined using unpaired, multiple *t* test with no correction for multiple comparison. Differentially abundant proteins and genes were analyzed using DAVID (Database for Annotation, Visualization, and Integrated Discovery, v6.8) to determine the biological processes and molecular functions associated with viral exposure in the presence or absence of ARV. Pathways with a minimum of at least two analytes associated and *p* < 0.05 were considered to be enriched.

## 3. Results

### 3.1. Expression of HIV-1 Receptor and Coreceptors in Colorectal Mucosa

The patterns of CD4 and CCR5/CXCR4 expression in human vaginal and small intestine tissues have been described previously and compared [27,28,29,30,31,32]. However, limited phenotype characterization has been performed with colorectal tissue, known to be highly permissive to HIV-1 [22,33,34,35]. Flow cytometry analysis of enzyme-digested colorectal tissue showed, as expected, a high level of CD4+ cells (60.73 ± 13.22%) and 35.93 ± 13.02% of CXCR4+ cells (Figure 1a). Due to the presence of different conformational forms of CCR5 on the surface of HIV-1 target cells [36] we used three MAbs (2D7, 45,531 and 45,523) targeting different conformational epitopes [37] to better assess the level of CCR5 in colorectal tissue. Furthermore, these MAbs have shown no cross-reactivity in CCR5-∆32 homozygous PBMCs [24]. CCR5 was detected on the cell surface with different levels of exposure for each epitope (8.29 ± 2.04% of 2D7^+^ cells, 40.56 ± 9.42% of 45531^+^ cells and 23.66 ± 19.07% of 45523^+^ cells) (Figure 1a). Similarly to previous analysis performed in vaginal and small intestine tissues [29], we measured the level of different cell population markers and innate response receptors. Compared to CD4^+^ cells, 17.20 ± 3.95% of cells were CD8^+^ cells and only 3.91 ± 2.01% of CD19^+^ cells were found in the digested samples. However, high levels of HLA-DR (45.95 ± 4.36%) and CD13 (60.00 ± 5.19%) were expressed in this tissue. Only 5.78 ± 0.81% of cells were DC-SIGN^+^ and the level of expression of innate response receptors was variable with 4.57 ± 4.23% of CD14^+^ cells, 16.73 ± 9.96% of CD64^+^ cells and 15.47 ± 10.62% of CD89^+^ cells. (Figure 1a). The expression of these receptors, as well as CCR5 and CXCR4, was similar to the levels detected within the CD4^+^ population (Figure 1b). Interestingly, 31.51 ± 10.08% of CD4^+^ cells were CD13^+^ (Figure 1b). In contrast to previous reports were CD13^+^ cells of small intestine did not express HIV-1 coreceptors [29], CD13^+^ cells isolated from colorectal tissue expressed CCR5 (3.21 ± 2.92% of 2D7^+^ cells, 42.54 ± 9.39% of 45531^+^ cells and 25.79 ± 8.05% of 45523^+^ cells) and CXCR4 (31.33 ± 11.35%) (Figure 1c).

We next assessed the morphological distribution of lymphocytes, dendritic cells (DCs) and macrophages in colorectal tissue by microscopy. Lymphocytes were mostly present in layer under the columnar epithelium (Appendix A) and some were found on the apical side of the epithelium (Appendix A). DCs were distributed in the lamina propia (Appendix A), across the epithelium (Appendix A) and even in the lumen (Appendix A). Macrophages were detected further in the lamina propia, underneath the layer of lymphocytes, at ≈44 µm from the lumen (Appendix A).

### 3.2. HIV-1 Penetration in Colorectal Explants

To visualize the interactions of HIV-1 with mucosal tissue occurring during in vivo transmission, colorectal explants were exposed to replication competent R5-tropic HIV-1 BaL (Appendix A). A nonpolarized system was chosen to include virus and lamina propia interactions that occur in microtears of the epithelium during RAI. Virions were labeled with photoactivatable GFP-Vpr (PA-GFP) to overcome confounding tissue autofluorescence and unequivocally identify individual virions within tissue sections. Colorectal explants were exposed to PA-GFP-HIV-1 BaL for 6 h allowing detection of virions following penetration in the explant and prior to viral budding. Virions were found in the layer of mucus, including colorectal secretory crypts (Figure 2a and bottom inset, respectively), and in contact with the surface of the columnar epithelium (Figure 2a top inset and Figure 2b). Viral particles were also detected across the epithelium (Figure 2c) and inside the lamina propia, up to 39 µm from the lumen, which corresponds to the layer of CD4+ lymphocytes described above (Figure 2d). After 6 h of incubation, no virions were detected in the deeper layer of macrophages. Photoactivatable signal was not detected in negative-control samples.

### 3.3. Early Responses Induced after Exposure of Colorectal Tissue to HIV-1

Early mucosal responses were evaluated 6 and 24 h post-ex vivo viral exposure. The time point of 6 h represents possible responses induced by viral attachment, entry and early reverse transcription. The 24 h point corresponds to viral integrations and productive infection. Ex vivo challenge with R5-tropic HIV-1 YU.2 or X4-HIV-1 NL4.3 induced a rapid (6 h post-challenge) and significant increase in the secretion of inflammatory cytokines, IFN-γ, chemokines, and GM-CSF (Figure 3a,b and Appendix A, Appendix A). At the later time point of 24 h, the increased expression of inflammatory cytokines, IFN-γ and chemokines was still detected (Figure 3b,c and Appendix A); however, less significant changes in the expression of cytokines/chemokines were observed in the infected explants compared to the unchallenged tissue than those observed at 6 h post-challenge.

Exposure to NL4.3 resulted in an increased secretion of more cytokines/chemokines and with greater fold changes observed than with YU.2. For example, the increase of MIP-1α induced by NL4.3 compared to unchallenged explants, was 2.1× at 6 h and 1.5× at 24 h higher than the increase observed with YU.2. Interestingly, the X4-tropic NL4.3 induced secretion of higher levels of monokine induced by IFN-γ (MIG) and CCR5 ligands (regulated upon activation, normal T cell expressed and presumably secreted (RANTES), MIP-1α and MIP-1β) in culture supernatant than the R5-tropic YU.2 (Appendix A).

The increased secretion of inflammatory cytokines, IFN-γ and chemokines such as IL-8 was not observed in colorectal explant cultures exposed to AT-2 inactivated virus (Appendix A).

Transcriptomic analysis of tissue explants confirmed the different mucosal proteomic modulations induced by NL4.3 and by YU.2 (Figure 4a,b, Appendix A). Mapping of gene interactions 6 h post-challenge (Figure 4c) identified two nodes centered around *IL-6* and *IFN-γ*. Interestingly, *IL-6* was downregulated upon viral infection. The upregulation of *IFN-γ* observed only reached statistical significance in explants infected ex vivo with NL4.3 (*p* = 0.011766) and not with YU.2 (*p* = 0.110011). Modulation of this node was connected to the axis of MIG (CXCL9), IP-10 (CXCL10) and interferon-inducible T-cell alpha chemoattractant (I-TAC, CXCL10) cytokines secreted in response to IFN-γ. Analysis of samples 24 h post-challenge (Figure 4d), revealed a number of key interaction nodes focused around IL1B, CCL5, CCR7, MMP9, PPARG, MMP3, NOS2A, and PTGS2. Along with minor nodes focused on CXCL9, CXCL10, CXCL11, MMP3, CCL3, CCL8, IFI44, and IL1RN. Greater relative downregulation of *IL1B* in YU.2 than in NL4.3 infected explants was observed compared to control unchallenged tissue. Modulation of *IL1B* positively correlated with regulation of *MMP3*, *MMP9*, *MMP12*, *PI3*, and *SERPINB2*. These same genes showed a range of responses to NL4.3 challenge and were either unchanged (*MMP3*), upregulated (*MMP12*) or downregulated (*SERPINB2*).

In parallel to the predominant transmission of R5-tropic HIV-1 through sexual intercourse [38,39], productive infection was observed only in tissue explants exposed to R5-tropic isolates and not to NL4.3 (Appendix A).

### 3.4. Modulation of Early Responses to HIV-1 Exposure by ARVs

Next, we evaluated the effect of ARV candidates for pre-exposure prophylaxis (PrEP) on the mucosal responses induced by HIV-1 infection. ARVs with mechanisms of action targeting early steps in the viral replication cycle were chosen: tenofovir (TFV), a nucleotide reverse transcriptase inhibitor (NRTI); UC-781 and dapivirine (TMC120), two nonnucleoside RTI (NNRTI); T1249, a fusion inhibitor; AMD3465, an X4-tropic virus entry inhibitor; CMPD167, an R5-tropic virus entry inhibitor. Known ex vivo HIV-inhibitory concentrations in colorectal explants were used [25,40] and confirmed for this study in tissue and with in vitro assays (Appendix A). None of the ARVs tested significantly modulated the mucosal cytokine profile of unchallenged colorectal explants (Appendix A). Six hours after dosing and challenge of colorectal explants (Figure 5a,b), all ARVs tested induced downregulation of IL-1α that had been upregulated after challenge with YU.2 or NL4.3. IL-2 was significantly downregulated by all ARVs tested prior to challenge with NL4.3 but upregulated when colorectal explants were challenged with YU.2 (Appendix A). However, the concentration of most cytokines that were upregulated following challenge in the absence of ARV, were further increased by the ARVs tested (Appendix A).

At the later time point of 24 h (Figure 5c,d), dosing of explants prior to viral challenge induced a downregulation of IL-8, MCP-1, IL-6, G-CSF, and IL-1α which were upregulated in infected explants and at the earlier time point of 6 h in explants dosed with ARV. Modulation of other cytokines was ARV-dependent and distinct for explants challenged with NL4.3 or YU.2. Dosing with ARVs, induced a significant reduction of IL-2 levels in explant cultures challenged with NL4.3, but a significant upregulation with YU.2. TGF-β was significantly upregulated by all the ARVs when tested against NL4.3, however, upregulation was only induced by AMD3465 (a CXCR4 ligand) against R5-tropic YU.2 (Appendix A).

## 4. Discussion

The systemic cytokine/chemokine profile induced during the acute phase of HIV-1 infection has been well characterized [5,6]; however, the early mucosal responses to viral exposure during sexual transmission are still not fully understood and have mainly been analyzed using primary cells isolated from the female or male reproductive tracts and from the colorectum [41,42,43]; or in stimulated tissue models [44].

Our CCR5 staining strategy allowed us to characterize the expression of this coreceptor in colorectal tissue and revealed how both CD4^+^ and CD13^+^ cells express similar levels of CCR5 and CXCR4. This is in contrast with what has been described for small intestine macrophages which express very low levels of CD4, and both HIV-1 coreceptors [28,29]. Jejunal DCs express high levels of CD4 and coreceptors [30]. In the lower FGT, CD4^+^ T cells and macrophages express both coreceptors, but Langerhans cells are CXCR4^−^ and DC-SIGN^−^ [29,32,45,46]. However, the expression level of CCR5 in vaginal macrophages remains unclear [29,45]. Interestingly, endometrial and endocervical CD4^+^ T cells express higher levels of CCR5 but less CXCR4 than T cells from the lower FGT [31].

In this study, we used the ex vivo challenge model of unstimulated mucosal tissue explants to measure the effect of HIV-1 transmission in the colorectal tract within the first day of exposure. After 6 h of inoculation, we already found an increased secretion in colorectal explant culture supernatants of inflammatory cytokines, chemokines, and IFN-γ. The inflammatory and Th1 cytokine responses was observed with both clade B isolates tested independently of their tropism. However, X4-tropic HIV-1 NL4.3 induced greater modulation and affected the expression of more cytokine/chemokines than the R5-tropic YU.2 isolate. These profiles were also observed at the transcriptomic level. Interestingly, the secretion level of IFN-γ (MIG) and CCR5 ligands (RANTES, MIP-1α and MIP-1β) was further upregulated in response to NL4.3 than to YU.2. Ito et al. have described the same effect in human tonsil explants [47]. The greater innate mucosal responses to X4-isolates exposure compared to R5 could be responsible for the predominant transmission of R5-tropic HIV-1 isolates through sexual intercourse [38,39]. The different cytokine profiles induced by exposure to R5 and X4 tropic isolates could be due not only to the tropism of the virus, but also to factors such as the signaling resulting from the binding of gp120 to CCR5 or CXCR4 [48] and potential differences in the size of the founder population. Several reports have described rectal transmission of multiple founder variants [49,50,51] compared to the majority of single founder transmissions in the FGT [50,52].

The inflammatory profile induced in colorectal tissue following exposure to HIV-1 has also been described in the FGT with increased levels of IL-6, IL-10, IL-12p70, IL-8, and IL-1β in cervicovaginal lavages from women during acute HIV infection [53].

Microscopy analysis of ex vivo challenged colorectal tissue revealed that during the first 6 h of R5-tropic HIV-1 BaL transmission, virions remained in the layer of CD4^+^ cells beneath the columnar epithelium and did not migrate deeper into the layer of macrophages (Figure 2 and Appendix A) despite the high macrophage replication capacity (MRC) of this isolate compared to primary isolates in in vitro cellular assays [54].

TFV has been among the first ARVs approved by the FDA for HIV-1 PrEP. However, ex vivo dosing with TFV of primary cells from the FGT has shown upregulation of inflammatory cytokines [41]. In vivo rectal dosing with TFV downregulated mucosal anti-inflammatory factors [55,56]. Here we did not observe any innate response after exposure of colorectal explants to a range of ARVs with different mechanism of action, including RTIs, fusion and entry inhibitors; however, when used prior to viral challenge, the mucosal profiles induced by NL4.3 and YU.2 were affected by the different compounds and the effect was in some cases opposite between 6 h (inflammatory) and 24 h (anti-inflammatory) post-challenge and dependent on the viral isolate used for challenge.

The main limitation of this study is the use of laboratory-adapted strains instead of transmitted founder isolates. The use of clonal isolates allowed a better characterization of the responses to each strain without the presence of quasi species for this proof-of-concept study. Furthermore, viral isolates were not purified after passage in PBCMs and therefore could contain additional factors other than the virus that might have modulated the responses; however, the viral stocks for both viruses, YU.2 and NL4.3, were grown at the same time with PBMCs from the same donors. Another limitation is that despite, the high MRC of both, YU.2 and BaL, mucosal profile in response to HIV-1 BaL was not evaluated. Our study was not set up to detect neither the first cells targeted by the virus, nor the number of cells productively infected nor to visualize the effect of ARVs on the cells in foci of infection. Future studies will also be necessary to identify the cellular populations responsible for the early responses and their modulation during the first hours of viral exposure.

This study demonstrates that the inflammatory and Th1 profiles of the acute phase of HIV-1 infection are elicited in the first 24 h of transmission in the colorectal site of entry, which correspond to the events prior to viral dissemination to draining lymph nodes.

Human studies are limited by the availability of mucosal samples from HIV-infected patients and by the estimation of the infection time point. We have previously shown that in vivo viral replication fitness can be mimic in mucosal tissue explants [25]. This study further supports the use of ex vivo mucosal challenge models as surrogates of in vivo transmission. Hence, the tissue explant model represents a valuable tool to investigate the responses to viral exposure or to prevention strategies within the context of the mucosal compartment, for the design of effective prevention strategies including vaccines.

## Figures and Tables

**Figure 1 vaccines-09-00231-f001:**
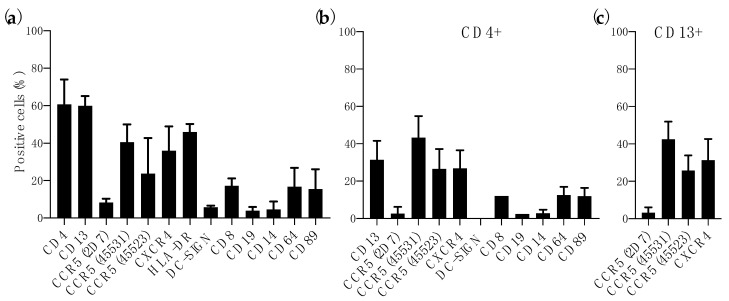
Characterization of immune cells within human colorectal tissue. Percentage expression of immune cell markers and/or HIV-1 receptor and coreceptor in (**a**) total colorectal cells, (**b**) CD4^+^ cells and (**c**) CD13^+^. Data shown are mean/SD of at least three independent experiments performed with tissue from different donors.

**Figure 2 vaccines-09-00231-f002:**
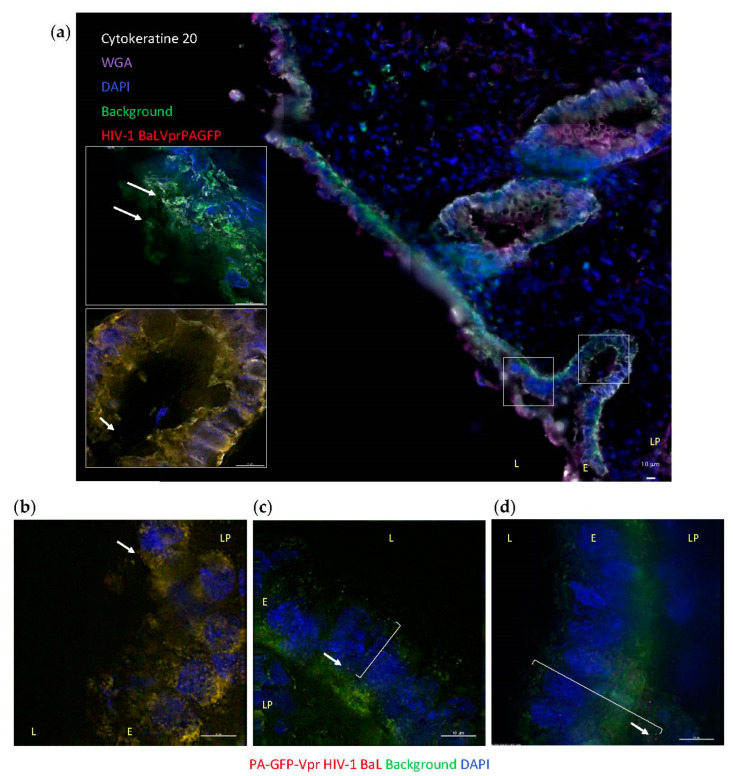
Ex vivo HIV-1 interactions with human colorectal tissue. Colorectal explants were exposed to PA-GFP-Vpr HIV-1 BaL (red and signaled with arrows) for 6 h. (**a**) Representative image of cryosections stained with cytokeratine 20-RedX (white) and WGA-Cy5 (purple) to label the columnar epithelium; autofluorescence is shown in green; top and bottom insets correspond to left and right boxed areas in main image, respectively. (**b**–**d**) Representative images of cryosections with autofluorescence shown in green; brackets indicate distance of virion from the tissue surface: (**c**) 18.6 μm and (**d**) 39 μm. Cell nuclei stained with DAPI (blue). White bar = 10 μm. L: lumen; E: epithelium; LP: lamina propia.

**Figure 3 vaccines-09-00231-f003:**
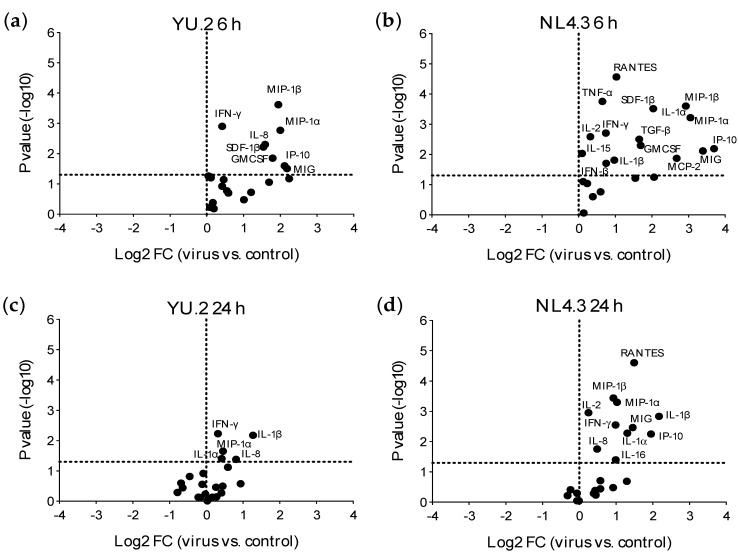
Modulation of colorectal cytokine profile induced by exposure to HIV-1. Volcano plots of all cytokines measured after 6 (**a**,**b**) or 24 h (**c**,**d**) of culture and comparing colorectal explants exposed to YU.2 (**a**,**c**) or NL4.3 (**b**,**d**) to those unchallenged (control). FC: fold change. Horizontal dotted line indicates a *p* value of 0.05. Analytes above this line are significantly modulated. Vertical dotted line is set at a Log_2_FC = 0. Data shown are from four independent experiments performed with tissue from different donors.

**Figure 4 vaccines-09-00231-f004:**
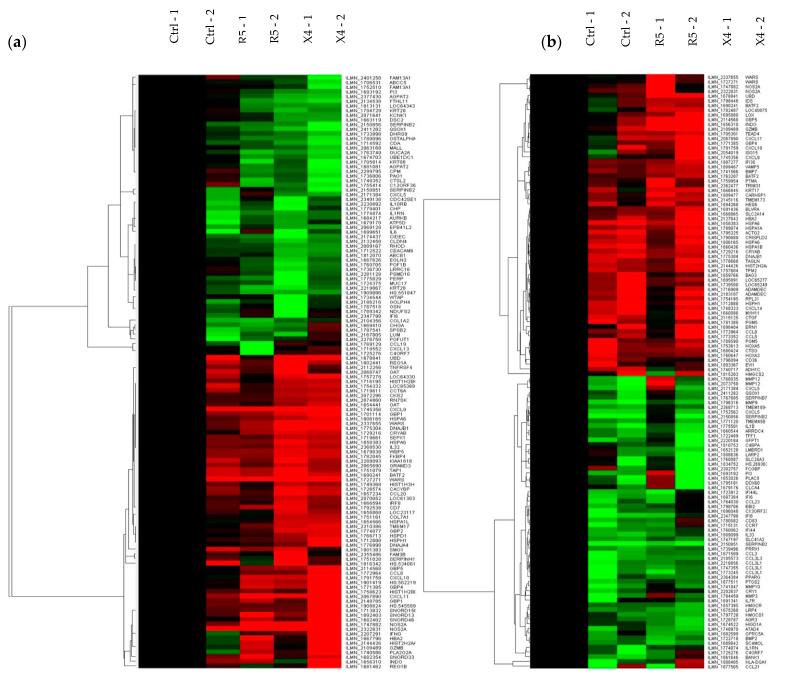
Transcriptomic analysis of HIV-1 YU.2 or NL4.3 ex vivo challenge of colorectal explants. RNA-microarray data from colorectal explants cultured (**a**) 6 or (**b**) 24 h post-ex vivo challenge was averaged by treatment and filtered on 1.5-fold change relative to the control. Hierarchical clustering was performed on data prior to averaging using uncentered correlation and average linkage. Columns represent individual samples and rows genes that are up (red) or downregulated (green). Interaction maps on genes across (**c**) 6 and (**d**) 24 h time points with a fold change greater than 1.5 in explants infected with either NL4.3 or YU.2 compared to untreated. The color swatches above each node indicate relative expression for the averaged treatment i.e., none (control), R5 (YU.2), X4 (NL4.3) left to right. Blue: downregulated; red: upregulated; yellow: unchanged. Data are shown from two independent experiments.

**Figure 5 vaccines-09-00231-f005:**
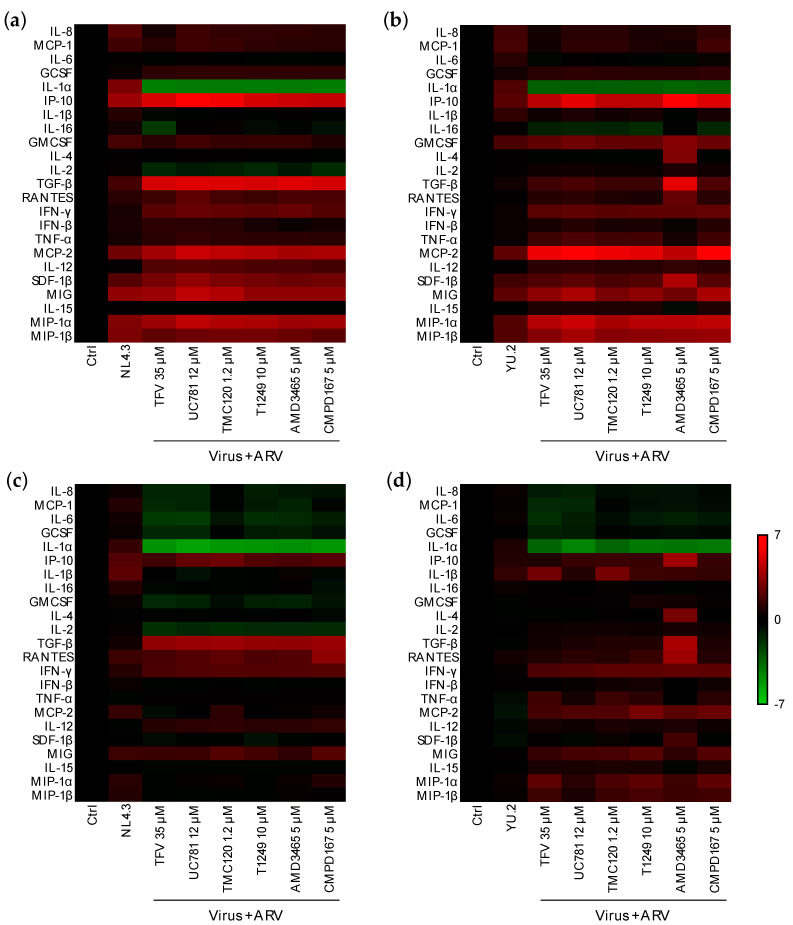
Effect of ARVs on the mucosal cytokine profile induced post-viral challenge of colorectal explants. Heatmap representing cytokines that are upregulated (red) or downregulated (green) (**a**,**b**) 6 or (**c**,**d**) 24 h after challenge in the presence or absence of ARV in comparison to untreated explants. Differences are shown in Log2 from two independent experiments performed in quadruplicate.

## Data Availability

Data available upon request.

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
