# Peer review of "Early Colorectal Responses to HIV-1 and Modulation by Antiretroviral Drugs"

_vaccines, 2021, doi:10.3390/vaccines9030231_

Round 1

Reviewer 1 Report

The authors investigated the early cellular responses to an ex-vivo HIV-1 infection in colorectal tissues. They showed by microscopy that the virus can penetrate the tissues in the 6 hours after inoculation. Experiments with two HIV-1 isolates (R5- and X4-tropic) shows, with some differences between the isolates, an increase in secretion of inflammatory cytokines, chemokines, IFN-g, and GM-CSF after inoculation. These results were also confirmed at the transcriptomics level with microarrays. The authors then tested the effects of pre-treating potential antiretroviral (ARV) drugs candidates for colorectal microbicides before HIV-1 inoculation on the early cellular responses and observed some modulations in the responses.

Overall, while it is an interesting manuscript which should be of interest to the community, there is some controls missing limiting the interpretations of the data.

Comments:

  1. The authors should discuss whether the early responses they observed in colorectal tissues are similar to what have been observed in vaginal or small intestine tissues (immune cells/HIV-1 receptor/co-receptors)?
  2. While the manuscript shows by microscopy shows that HIV1 can penetrate in colorectal tissues, a major limitation is the absence of data showing the proportion of cells infected by HIV-1. It remains unclear for example whether YU.2 and NL4.3 infect a similar number of cells or what happened to HIV-1 infection when the colorectal tissues are pre-treated with ARV drugs. This information is important to be able to conclude about the effect of the HIV-1 isolates or of the ARV drugs on the early cellular responses (see points 3 and 4 below).
  3. Can the differences in the secretion of inflammatory cytokines, chemokines, and IFN-g between the two HIV-1 isolates could (also) be explained by the differences in number of cells in the tissues with CCR5 or CXCR4? More cells infected leading to a higher secretion of inflammatory molecules?
  4. While the authors show that the ARV drugs do not affect secretion of inflammatory molecules, pre-treatment of colorectal tissues with ARV drugs before HIV-1 inoculation show a wide-range of change (compared to inoculation alone) in the secretion of inflammatory molecules (down-regulation of some molecules but up-regulation of others). As there are no data on the viral load or number of cells infected by HIV-1 in presence of the different ARV drugs, it is complicated to make sense of the modulations on the early responses made by the ARV drugs (and whether the ARV drugs has any effects on HIV-1).

Author Response

Response to Reviewer 1 Comments

Overall, while it is an interesting manuscript which should be of interest to the community, there is some controls missing limiting the interpretations of the data.

We agree that controls should be included, and we have now added data available of the cytokine profile induced by inactivated YU.2 and NL4.3 which could only be performed with one donor due to the size of the tissue available (lines 298-300 and Figure S4). The amount of tissue was a limiting factor to include further controls in our studies. We have also confirmed that no signal corresponding to PA-GFP was detected in negative-control samples (lines 261 -262).

Comments:

  1. The authors should discuss whether the early responses they observed in colorectal tissues are similar to what have been observed in vaginal or small intestine tissues (immune cells/HIV-1 receptor/co-receptors)?

We agree that these aspects are important to comment in the discussion. We have compared the expression level of CD4 and HIV co-receptors in the female genital tract and in the small intestine (lines 357 – 366) and the data available of cytokine responses in the female genital tract during acute infection. (lines 386 – 388).

  1. While the manuscript shows by microscopy shows that HIV1 can penetrate in colorectal tissues, a major limitation is the absence of data showing the proportion of cells infected by HIV-1. It remains unclear for example whether YU.2 and NL4.3 infect a similar number of cells or what happened to HIV-1 infection when the colorectal tissues are pre-treated with ARV drugs. This information is important to be able to conclude about the effect of the HIV-1 isolates or of the ARV drugs on the early cellular responses (see points 3 and 4 below).

The microscopy studies were design to assess virion penetration in tissue, however no p24 staining was performed to detect the initial focus or potential foci of infection. We acknowledge that this study was not set up to detect the first infected cell in the first hours of challenge, to determine the number of cells productively infected by any of the viral isolates nor to visualize the effect of ARVs on the founder population, which would require staining of tissue explants with anti-p24 and at different time points. We have commented this limitation in the discussion (lines 393 – 395). We have added data of viral replication kinetics of the three viral isolates used in the study (lines 249, 316-318 and Figure S2) and ARV activity from explant cultures set up in parallel for some donors when sufficient tissue was available (lines 326-327 and Figure S7). The activity of ARVs was only tested against the R5-tropic YU.2 due to the lack of productive infection of X4-tropic isolates in colorectal tissue. To demonstrate the activity of AMD3465, an X4-tropic entry inhibitor, we have added the activity profile in TZM-bl cells in the supplementary Figure S7.

  1. Can the differences in the secretion of inflammatory cytokines, chemokines, and IFN-g between the two HIV-1 isolates could (also) be explained by the differences in number of cells in the tissues with CCR5 or CXCR4? More cells infected leading to a higher secretion of inflammatory molecules?

We thank the reviewer for highlighting this point to improve the discussion. We agree that several factors could be involved in the modulation of the mucosal cytokine profile including the number of cells infected. Several reports have shown increased transmission of multiple HIV variants in the colorectal tract compared to the female genital compartment. We have added a section in the discussion (lines 369-375).

  1. While the authors show that the ARV drugs do not affect secretion of inflammatory molecules, pre-treatment of colorectal tissues with ARV drugs before HIV-1 inoculation show a wide-range of change (compared to inoculation alone) in the secretion of inflammatory molecules (down-regulation of some molecules but up-regulation of others). As there are no data on the viral load or number of cells infected by HIV-1 in presence of the different ARV drugs, it is complicated to make sense of the modulations on the early responses made by the ARV drugs (and whether the ARV drugs has any effects on HIV-1).

We agree that there are technical limitations that do not allow us to fully characterize aspects such as number of cells infected within 24 h of viral exposure or the effect of ARVs on these cells; however, we did, when sufficient tissue was available, confirmed the activity of the ARVs in tissue explants and TZM-bl cells (lines 326-327 and Figure S7) as mentioned in our reply to comment 2.

Reviewer 2 Report

The manuscript “Early Colorectal Responses to HIV-1 and Modulation by antiretroviral Drugs” by Herrera et al. is a research article aiming at evaluating the effect of HIV-1 transmission in the colorectal tract during the first 24 hours of exposure assessed in an ex vivo challenge model of unstimulated mucosal tissue explants.

General comment:

The subject addressed in this article is worthy of investigation, as clearly exposed in the introduction section.

Minor issues:

  1. Legibility of text in figures (such as figures 4, 5, supplementary figures 5) should be improved.
  2. Section 3.3: “greater fold change” ,  “secretion of higher levels”  ,  “confirmed greater modulation”  ,  “greater up-regulation”  ,  “greater relative down-regulation”. More qualitative comparison should be provided.
  3. Briefly mention additional information (in addition to relative reference) on photoactivatable GFP-Gag.

Author Response

Response to Reviewer 2 Comments

  1. Legibility of text in figures (such as figures 4, 5, supplementary figures 5) should be improved.

Thank you for this comment that will improve the quality of the figures. The size of legends has been increased as much as possible although for some figures this was not possible, however, we could submit separated files as supplementary material of individual components of Figure 4 with increased size if necessary.

  1. Section 3.3: “greater fold change” ,  “secretion of higher levels”  ,  “confirmed greater modulation”  ,  “greater up-regulation”  ,  “greater relative down-regulation”. More qualitative comparison should be provided.

We agree that the language could be improved and thank the reviewer for the comment. We hope that this section reads better (lines 284 – 286, 302 – 303, 306 – 308).

  1. Briefly mention additional information (in addition to relative reference) on photoactivatable GFP-Gag.

We agree that further detail will help the reader. We have added further details about the production of the fluorescent virus and the original reference of the PA-GFP in section 2.2 lines 78-82. Furthermore, we would like to apologize for a typo in the originally submitted manuscript were instead of Vpr we wrote Gag. This has been amended in the manuscript.

Round 2

Reviewer 1 Report

The authors have satisfactorily responded to all my questions and made the necessary changes to the manuscript.

Author Response

Thank you for your feed-back.